# Deep Regression Prediction of Rheological Properties of SIS-Modified Asphalt Binders

**DOI:** 10.3390/ma13245738

**Published:** 2020-12-16

**Authors:** Bongjun Ji, Soon-Jae Lee, Mithil Mazumder, Moon-Sup Lee, Hyun Hwan Kim

**Affiliations:** 1Department of Industrial and Management Engineering, Pohang University of Science and Technology, Pohang 37673, Korea; bongjun_ji@postech.ac.kr; 2Department of Engineering Technology, Texas State University, San Marcos, TX 78666, USA; SL31@txstate.edu (S.-J.L.); m_m624@txstate.edu (M.M.); 3Korea Institute of Civil Engineering and Building Technology, Goyang 10223, Korea

**Keywords:** deep learning model, regression architecture, atomic force microscopy, styrene–isoprene–styrene, dynamic shear rheometer

## Abstract

The engineering properties of asphalt binders depend on the types and amounts of additives. However, measuring engineering properties is time-consuming, requires technical expertise, specialized equipment, and effort. This study develops a deep regression model for predicting the engineering property of asphalt binders based on analysis of atomic force microscopy (AFM) image analysis to test the feasibility of replacing traditional measuring estimate techniques. The base asphalt binder PG 64-22 and styrene–isoprene–styrene (SIS) modifier were blended with four different polymer additive contents (0%, 5%, 10%, and 15%) and then tested with a dynamic shear rheometer (DSR) to evaluate the rheological data, which indicate the rutting properties of the asphalt binders. Different deep regression models are trained for predicting engineering property using AFM images of SIS binders. The mean absolute percentage error is decisive for the selection of the best deep regression architecture. This study’s results indicate the deep regression architecture is found to be effective in predicting the G*/sin *δ* value after the training and validation process. The deep regression model can be an alternative way to measure the asphalt binder’s engineering property quickly. This study would encourage applying a deep regression model for predicting the engineering properties of the asphalt binder.

## 1. Introduction

Even though bitumen has been widely used as a binder material in asphalt based on its viscoelastic properties [1,2,3], the use of the unmodified asphalt binder has several weaknesses such as low resistance on rutting and cracking. The addition of the polymer leads to a greater resistance to rutting and cracking by improving the elasticity of the material. There are many polymer additives in the asphalt pavement industry.

One of them is styrene–isoprene–styrene (SIS) polymer, which has excellent aging resistance, mixing stability and improved elastic responsiveness, cohesion, tensile strength, and low-temperature flexibility. The SIS molecule chain is composed of isoprene. It increases the complex modulus at high-temperature, and due to its branch methyl in the isoprene group, it has better tenacity and compatibility with other materials. As a result, SIS is used to mix with asphalt materials due to their high stiffness and elasticity to become polymer-modified asphalt (PMA) binder. According to the previous research [4], SIS has the positive effect of increasing the complex modulus at high-temperature by the molecule chain of isoprene. In addition, it has better tenacity and compatibility with other materials due to its branch methyl in the isoprene group. In fact, the addition of SIS into the binder showed improved performance compared to the control binders [5]. In addition to SIS, various additives have been investigated, including clay, fumed silica, nano-SiO_2_, and rubber powder. Ref. [6] investigated the composite of clay and fumed silica nanoparticles. They exposed the composite to ultraviolet light to lower stiffness and increase resistance to permanent deformation. Ref. [7] investigated the effect of nano-SiO_2_ and rubber powder as additives in asphalt binders. They found that the nano-SiO_2_ increase of stiffness modulus and rubber powder increases the elasticity and decreases the stiffness. Despite various compounds of asphalt, SIS still seems to be a promising additive.

With the application of modern technology, researchers have been able to inspect asphalt on a small scale using modern equipment such as atomic force microscopy (AFM) [8,9,10,11,12]. The captured images with AFM can reveal the heterogeneous domains in asphalt binders with different mechanical properties [13]. Several studies have evaluated the “bee-like” appearance used to characterize asphalt surfaces with asphaltenes [5,8,9]. In addition, the size and number of bee-like shapes are considered to reflect the asphalt binder property. Generally, the mechanical performances of asphalt binders are evaluated by Superpave binder tests. According to previous research [14], the microstructural analysis through AFM images is expected to provide insight into the binder performance. Application of AFM to investigate the microstructures of bituminous material is relatively new and a growing field. In order to come up with a reasonable conclusion regarding the micromorphology obtained from these emerging diagnostic tools, a larger body of knowledge and observations is required.

Even though the application of advanced technology positively supports the analysis of the binder surface at a very small scale, there is not a standardized evaluation method, which can bring the result without personal prejudice. The deep learning-based image evaluation method seems a promising tool to automatically evaluate and analyze the binder surface on a microscopic scale. Deep learning has been successfully applied to many images related to the applications such as image classification [15,16,17], object detection [18,19,20], and image regression [21,22,23]. The advantage of using the deep learning model is that it can be improved when more data are available [24]. They have outperformed not only traditional computer vision tasks but also human capacities in some cases [25,26]. Furthermore, the performance of the deep-learning model is consistent; otherwise, the performance of human experts may be variable. If the deep learning model is developed and applied to rheological properties prediction, the developed model can help the decision-making process of qualified experts along with enhancing the abilities of practitioners.

This study was conducted to develop the deep regression model for predicting the engineering property of asphalt binders based on AFM image analysis. The deep regression model is a type of deep learning model used to predict continuous numbers. The SIS-modified binders were produced in the laboratory with four different polymer additive contents, 0%, 5%, 10%, and 15%. A dynamic shear rheometer (DSR) test was performed to provide the rheological data, which indicate the rutting properties of the binders. Basically, the asphalt rheology can be examined by measuring two viscoelastic parameters to predict the behavior of pavement condition, including complex modulus (G, stiffness) and phase angle (*δ*, viscous or elastic index) for various test temperatures and frequencies. Therefore, the higher G*/sin *δ* values indicate that the binders will be less susceptible to permanent deformation at high pavement temperatures by a stiff portion that resists the permanent deformation. Each asphalt binder is tested to determine the G*/sin *δ*.

Topographic images of four PMA binders were captured by AFM and analyzed considering specific patterns and shapes on the surface. AFM images were preprocessed before training with eight deep learning architectures (LeNet-5, AlexNet, VGG-16, ResNet, LeNet-5_LR, AlexNet_LR, VGG-16_LR, and ResNet_LR). Most of the architecture is designed for classification problems that predict categorical values rather than regression problems that predict continuous values. However, the problem we dealt with in this paper is a regression problem. Therefore, we have tested the existing popular architecture itself and modified the architecture to fit the regression with a little tuning. After training and validation, all prediction results of architectures were evaluated with three regression performance metrics; R-Squared, mean squared error (MSE) and mean absolute percentage error (MAPE). Figure 1 shows a flow chart of the experimental design used in this study.

## 2. Experimental Design

### 2.1. SIS-Modified Binder Production

Performance grade (PG) 64-22 asphalt binder was used to produce the styrene isoprene styrene (SIS) binder. The binder properties are presented in Table 1. The SIS binder was produced in the laboratory at 180 °C for 80 min by an open blade mixer at a blending speed of 700 rpm. The percentages of SIS were 0%, 5%, 10%, and 15% by weight of the base binder. One batch of SIS additive was used in this study to have the consistency of the SIS binders throughout the study.

### 2.2. Dynamic Shear Rheometer (DSR) Test

The high-temperature rheological properties of SIS binders are measured using a dynamic shear rheometer (DSR) per AASHTO T 315. In the DSR test, the original binder is tested with a 25 mm spindle at 76 °C and a parallel plate. Figure 2 shows a DSR testing apparatus used in this study. In the DSR test, the binders are tested at a frequency of 10 radians per second, which is equal to approximately 1.59 Hz.

### 2.3. Atomic Force Microscope (AFM)

The 840-002-380 Bruker Dimension Icon AFM model (Bruker Instrument Inc. Billerica, MA, USA) was used to characterize the micromorphology of SIS binders through the surface images obtained on a prepared sample. This AFM is shown in Figure 3. AFM can be used to measure the forces between the tip and the sample as a function of their mutual separation. The AFM tapping mode imaging was performed on the binder samples to evaluate the morphology of the binder. In the tapping mode, the AFM tip was oscillated at its resonance frequency by a piezoelectric element connected to the tip holder assembly. The piezo-drive was regulated using feedback control to keep a constant tip-upsample distance (setpoint) (Bhushan and Qi 2003). Topography images were obtained with a scan rate of 0.99 Hz and a scan size of 20 μm × 20 μm. Topography images represent the topography of the surface. Figure 4 shows the AFM equipment used in this study.

### 2.4. Deep Learning

The advantage of applying deep learning is that it excludes subjective views and provides results for objects. Deep learning as a technique itself has some limitations, such as requiring a large dataset, skilled experts for tuning hyper-parameter. However, the advantages outweigh the disadvantages in many cases. The purpose of this study was to find out whether it was possible to predict the performance of a binder only by observing the microscopic images rather than obtaining the engineering characteristics through lab experiments. Figure 4 shows the flow chart for the deep learning process.

#### 2.4.1. Data Matching

Image data (x) and property data (y) should be matched to train the prediction model. Figure 5 shows the conceptual diagram of data matching. Image data can be represented with a 3-dimensional array (x). The first two dimensions present the number of height and width, and the third dimension presents the three different color layers. For example, (2500, 2500, 3). The first two values match the number of pixels of the image; the third value represents the three colors: red, green and blue. Each row should be matched to measured property data G*/sin *δ* value (y).

#### 2.4.2. Data Preprocessing

It was required to conduct two data preprocessing before performing deep learning. First, preprocessing was the color scale adjustment. The equipment presents the height of the surface using a color scale. Each image was captured with a different color scale. The highest height was represented as white, and the lowest surface was represented as black. The problem was that each image had a different highest height. Therefore, the actaul height of the image can be vary depending on the image, even if the color was the same. To resolve this problem, adjusting the color to made the same color scale was essential before performing deep learning training. An example of the color-modified image is shown in Figure 6.

Another preprocessing was cropping. Collecting much image data had a limitation considering the laboratory environment. In general, predicting the rheological properties rely on a small portion of the image, such as specific patterns or shapes. Thus, the photos were cropped and then created to more images to be used for training data, validation data, and test data. The detailed process is illustrated in Figure 7.

#### 2.4.3. Data Augmentation

It is clear that much data can improve a model’s performance in deep learning. Data augmentation is the application of one or more variations to a collection of annotated training samples that generate new additional training data [27]. Data augmentation is widely used to overcome data shortage problems and reduce overfitting problems [24]. Examples of data augmentation technique and its result are shown in Figure 8.

#### 2.4.4. Train-Overview of Deep Learning Architecture

In the past decade, deep learning architecture in image processing has outperformed other traditional methods. The main deep learning architecture used for image processing is the convolutional neural network (CNN). CNN is a special type of neural network designed to recognize patterns in image data. CNN and its variants are now widely used for image processing, such as image classification (label prediction), object detection (detection of instances of semantic objects of a specific class), and regression (quantity prediction) problems. The most commonly mentioned in the literature are LeNet-5, AlexNet, VGG-16, ResNet. The LeNet-5 is the pioneering architecture of the deep learning model, and the remaining four architectures have earned a reputation for their performance in the ImageNet large-scale visual recognition challenge (ILSVRC).

LeNet-5 is a pioneering seven-stage convolutional network suggested by LeCun et al. (1998). It is developed to recognize 32 × 32 pixels, grayscale handwritten numeric images and has been applied by several banks. The structure of the LeNet-5 is shown in Figure 9. The input is the image of 0~9 with a size of 32 × 32, and its output is ten nodes. Each node represents the number 0~9. LeNet-5 consists of seven layers, which has two convolutional layers, two pooling layers, and three fully-connected layers.

AlexNet is one of the CNN-based architecture that became famous after winning the 2012 ILSVRC competition. AlexNet is named after Alex Krizhevsky, a researcher who proposed the structure. AlexNet is the first deep learning model that won the ILSVRC competition. The model showed a significant performance gap with the existing winning model. Many researchers have acknowledged the performance of deep learning models, and since 2013, most contestants have used them. The architecture of AlexNet is shown in Figure 10.

VGGNet is the runner-up at the 2014 ILSVRC competition. The name VGG comes from the affiliation of the researchers who developed this model; Visual Geometry Group at Oxford University. The model is widely used because it is much more concise than GoogLeNet, which is the winner of the 2014 ILSVRC competition. VGGNet consists of 5 blocks containing two or three convolution layers and a pooling layer. VGGNet uses relatively small convolution filters, and the depth increased to 11, 13, 16, 19. Up to 16 layers, the accuracy increased as the number of layers increased. However, when the layer is increased to 19, compared to the increased computation complexity, the improvement in accuracy was not great enough. Therefore, we used VGG-16, which has 16 layers, as shown in Figure 11.

ResNet is the model that won the 2015 ILSVRC competition. ResNet stands for residual networks and is a model that dramatically increases the depth of CNN to 152 layers, as shown in Figure 12. The driving force behind ResNet’s dramatic high-performance lies in a module called a residual block. The residual block adds a separate connection that passes the input x from the previous layer without performing a convolution operation or just one convolution operation called skip connection. The role of the skip connection is to help the optimizer to optimize parameters more easily. It allows the optimizer to find the best parameters for better CNN performance even as the depth of the layer deepens and the number of parameters that need to be trained increases. The residual block which has skip connection is widely accepted in the recent CNN model.

#### 2.4.5. Train-Applied Architecture (Deep Regression)

The famous CNN architecture mentioned above was applied as the base architecture. The CNN architecture was created to solve the classification problem. However, the problem with this study was related to regression. To solve the regression problem, the modified CNN architectures were applied to replacing the number of output and loss functions. Classification generated predicted labels, and regression applies to quantity predictions. The expected values in this study should be quantities. Therefore, the regression was performed in this study.

To change the output, the number of nodes in the last layer was replaced with one. For loss functions, “mean absolute percentage error” was used in regression analysis instead of other common loss functions such as mean square error, mean absolute error, and mean bias error. Scale-independent measures are more appropriate when the range of values is wide.

In addition, two architecture groups were used in this study. Group 1 was the existing architecture that had one float-type output value. Group 2 had an addition of a linear regression layer to the existing architecture (Figure 13). To identify this, group 2 from the original architecture “-LR”’ was named right after the original designation. For example, the LeNet5-LR model used LeNet5 as a base architecture and the linear regression layer was added to the output layer of LeNet5 architecture.

#### 2.4.6. Evaluation

In spite of the success of the deep learning model, the performance of the deep learning model varies on the training data and hyper-parameter of the model. Therefore, training deep learning models is generally not the end of the process. It should be evaluated and tuned if the performance is not sufficient. The evaluation of prediction resulted was conducted using three regression performance metrics, R-Squared, mean squared error (MSE) and mean absolute percentage error (MAPE).

## 3. Results and Discussions

### 3.1. Dynamic Shear Rheometer (DSR) Test

A high G*/sin *δ* value from the DSR test can be represented to have less susceptibility to permanent deformation known as rutting. Thus, the higher G*/sin *δ* results in a decreased rutting potential at asphalt pavement. All SIS-modified binders were tested to measure the G*/sin *δ* values at 76 °C. The results are shown in Figure 14. The addition of SIS resulted in higher G*/sin *δ* values compared to the 0% SIS binder. The higher the content of SIS, the better rutting performance. The 15% SIS binder showed the highest G*/sin *δ* values among all the binder types. It indicates that the addition of SIS is specifically effective in improving the rutting resistance of asphalt binders. These results can also be found with the correlation analysis between the G*/sin *δ* values and SIS contents. As mentioned, the high G*/sin *δ* values are considered desirable attributes in terms of resistance to plastic deformation of the pavements. It was found that the G*/sin *δ* values showed an obvious correlation with the SIS content (R^2^ value of 0.9942). The correlation results showed that the G*/sin *δ* values increased as the SIS content increased, suggesting that the higher SIS content increased the rutting resistance. This study uses the actual binder performance obtained through the DSR test for deep regression model training. In addition, the actual performance is used to compare and validate the predicted value by deep regression.

### 3.2. AFM Image Analysis

Figure 15 shows the topographic images of SIS-modified asphalt binders at a scan size of 20 µm. In Figure 15a, the topographic image of SIS 0% consists of a microstructure that looks like a sequence of hills and valleys known as a bee-like structure. The 0% SIS had an increased number of bee structure which was embedded on the surface compared to the images of other SIS contents. Figure 15b presents the topographic image of the asphalt binder containing the 5% SIS modifier. In the image, it was found that the number of bee structures was reduced compared to the 0% SIS. This result is more clearly shown in Figure 15c, which illustrates the topographic image of the 10% SIS. The addition of a further amount of SIS modifier decreased the size and width of the bee structure. According to the previous study [14], the microcrystalline waxes and waxy molecules were responsible for the bee structures. The addition of SIS modifiers was considered to contribute to the disappearance of bee-like structures and dissolves less rigid waxy molecules compared to control binders without SIS modifiers. The further addition of the SIS modifier up to 15% reduced the size and number of bee structures, as shown in Figure 15d. The trend that the number of bee structures in the topographic image decreased as the amount of modifier increased was consistent with earlier findings. This indicated that the addition of the SIS modifier contributed to the improvement of the rutting resistance of the binder by reducing the rigid portion having a bee-shaped structure. In addition, a new phase that had an oval shape started to appear in Figure 15c. These new phases identified in Figure 15d had brighter and smaller oval shapes that covered more areas rather than the image of SIS 10%. According to the DSR results, 15% SIS binder was evaluated to had the best performance with the highest G*/sin *δ* in terms of rutting resistance. This means that forming a new phase of oval shape at high SIS content affects the performance improvement of the basic binder PG 64-22, and this change could be observed through the micromorphology shown in the images of the PMA binders.

### 3.3. Prediction of Rheological Properties Using Deep Regression Model

The deep regression model was trained with DSR test results and AFM images. Figure 16 illustrated the training and validation loss during the training process of the eight deep regression architectures. The red line shows training loss, and the blue line indicates the validation loss overtraining process. If both the training and validation curves were very close to each other, go down together, it means that the model was trained well during training. It was expected that predicted values were close to the actual measured values. For example, Figure 16g showed minimal red color, indicating that the validation part hides almost all training parts. However, as shown in Figure 16d,h, there was a little overlap, indicating both ResNet50 and ResNet50+LR were not trained well and will not predict actual value well enough.

The average predicted values for each test set are shown in Table 2. Even though it did not predict accurately, most of them showed the right trend for predicting property value. If the actual value was small, a small value was predicted, and an image with a large measured value generally predicted a large value. This indicates that the trained model had the ability to identify features of the image, distinguish the difference based on features of the image and predict the value based on that difference. Not only the prediction but also the evaluation of prediction resulted was performed with the following regression performance metrics; R-Squared, mean squared error (MSE) and mean absolute percentage error (MAPE).

R^2^ (R-squared) is a measure of the proportion of the variance explained by an independent variable or variables in a regression model. Table 3 shows the R^2^ values of the prediction result for each architecture. In terms of R^2^ value, ResNet50 shows the best performance, followed by AlexNet and ResNet50_LR. All architectures with linear regression showed lower R^2^ values than normal architectures except for the LeNet5. It means that the linear regression model (LR model) does not guarantee better performance.

MSE measures the average of the squares of the error using the following Equation (1). The evaluation results of the MSE are shown in Table 4. According to the MSE results, ResNet-based architectures, including ResNet50_LR, showed the best performance. The MSE value where the actual value was large (10.25 in this study) had a great influence on the MSE average, indicating that it was scale-dependent. In this study, the actual smallest value of G*/sin *δ* was 0.51, and the actual largest value of G*/sin *δ* was 10.25. Since the smallest value and the largest value differ by around twenty times, it may lead to misinterpretation due to a scale difference.
(1)$$\mathrm{MSE}=\frac{1}{n}\sum_{i=1}^{n}{\left(Y_{i}-\hat{Y_{i}}\right)}^{2}$$

MAPE measures the percentage error of the predicted value with the following Equation (2), and the results are represented in Table 5. Unlike MSE, MAPE is scale-independent. Therefore, there were many differences compared to the MSE results. In R^2^ and MSE, ResNet50 showed the highest performance, but from the MAPE point of view, VGG16 showed the highest performance. VGG16 did not fit well with 10.25 and had a poor performance in R^2^ and MSE, but MAPE eliminated scale-dependency and showed the best performance.
(2)$$\mathrm{MAPE}=\frac{1}{n}\sum_{i=1}^{n}\left|\frac{Y_{i}-\hat{Y_{i}}}{Y_{i}}\right|\times 100$$

It turns out that there were no dominant architectures based on the three regression performance metrics. The used of a more complex and advanced algorithm did not guarantee better performance. MAPE was considered as the best performance metric based on the deep regression process in this study, including training and validation. In MAPE results, VGG16 was seemed to be the most effective architecture in predicting the G*/sin *δ*. Therefore, the prediction result of VGG16, which had the best performance from the MAPE perspective, was analyzed, and Figure 17 shows the results. A test set with an actual value of 0.51 predicts with almost 100% probability (Figure 17a). In Figure 17b, more than 55% of values were placed between 2.5 and 3. In addition, about 20% of values were distributed between 0.5 and 1. It was thought to had been confused with an image having a value of 0.51. As this phenomenon appeared in the rest of the test set (6.61 and 10.25), it was shown that the developed model tended to confuse adjacent values with actual values. The tendency of the developed model to confuse predicted values with adjacent values based on AFM images was also evidence that the model recognizes and distinguishes the microstructural characteristics of the image. This indicates that the model could identify the correlation between the AFM images and G*/sin *δ* values, showing binder performance.

## 4. Summary and Conclusions

Deep regression was applied to evaluate the engineering properties of SIS-modified asphalt binders based on the AFM image analysis captured at a scan size of 20 µm. AFM images were processed and then trained, considering the G*/sin *δ* values measured by the DSR test. Based on the results of this study, the following summary points can be drawn:The addition of SIS into asphalt binder could significantly increase the G*/sin *δ* at a high-temperature, which could ensure better rutting resistance of asphalt binder;The microstructural changes such as bee-like structure and oval shape depending on SIS contents were observed on AFM images, and these properties correlate with the rutting resistance measured by the DSR test;In the training and validation process, different training results were presented depending on deep regression architectures, indicating that the type of architecture affects prediction performance on the rheological value measured by the DSR test;Even though the linear regression layer was added to the existing architecture considering the quantity prediction, it was concluded that the linear regression models (LR model) could not guarantee a better performance to predict the value;Mean absolute percentage error (MAPE) was considered to have the best performance for the evaluation of prediction results in this study;Based on the prediction result of VGG16, which placed the highest rank from the MAPE perspective, it was found that the predicted value generated by the developed model showed confusion between the actual value and the adjacent value. This tendency paradoxically indicated the ability of the model to identify the relation between the AFM images and G*/sin *δ* values related to the binder performance;Future work will focus on the development of a more general deep regression model for predicting G*/sin *δ* according to the change of additive types.

This study is considered as first research that uses a deep regression model on AFM image to predict G*/sin *δ* values. The developed model predicted G*/sin *δ* among the various engineering properties. The finding of our study indicates that images can reveal the engineering properties of asphalt binder, and a deep regression model can train the relation between images and engineering properties. If other engineering properties can also be predicted through images, a deep regression model can replace various experiments and equipment to measure various engineering properties. The use of a deep regression model can save the time and cost required for the current measuring technique. This study serves as a stepping stone for future research to further develop deep regression models to predict the engineering properties of asphalt binders.

## Figures and Tables

**Figure 1 materials-13-05738-f001:**
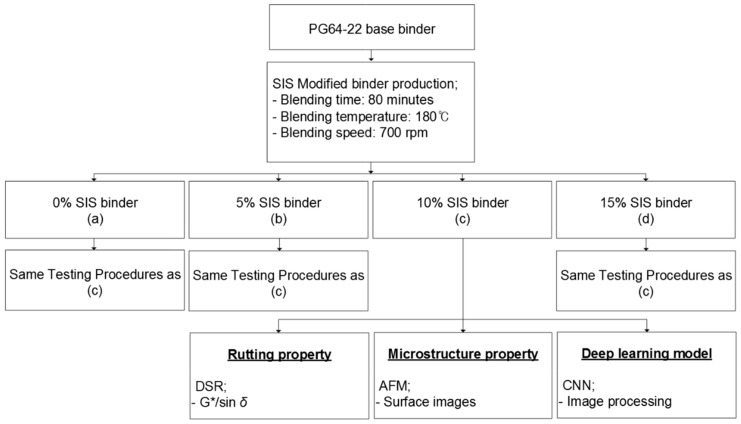
Flow chart of experimental design procedures.

**Figure 2 materials-13-05738-f002:**
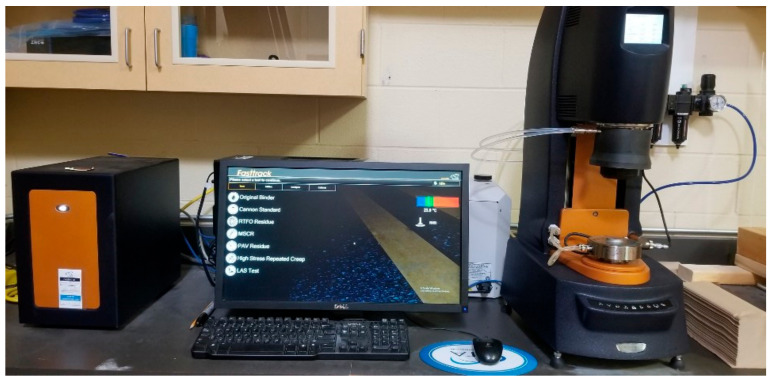
Dynamic shear rheometer (DSR) tester.

**Figure 3 materials-13-05738-f003:**
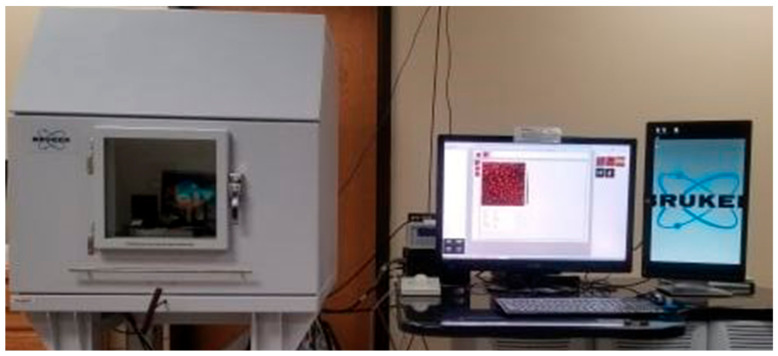
Atomic force microscopy (AFM).

**Figure 4 materials-13-05738-f004:**
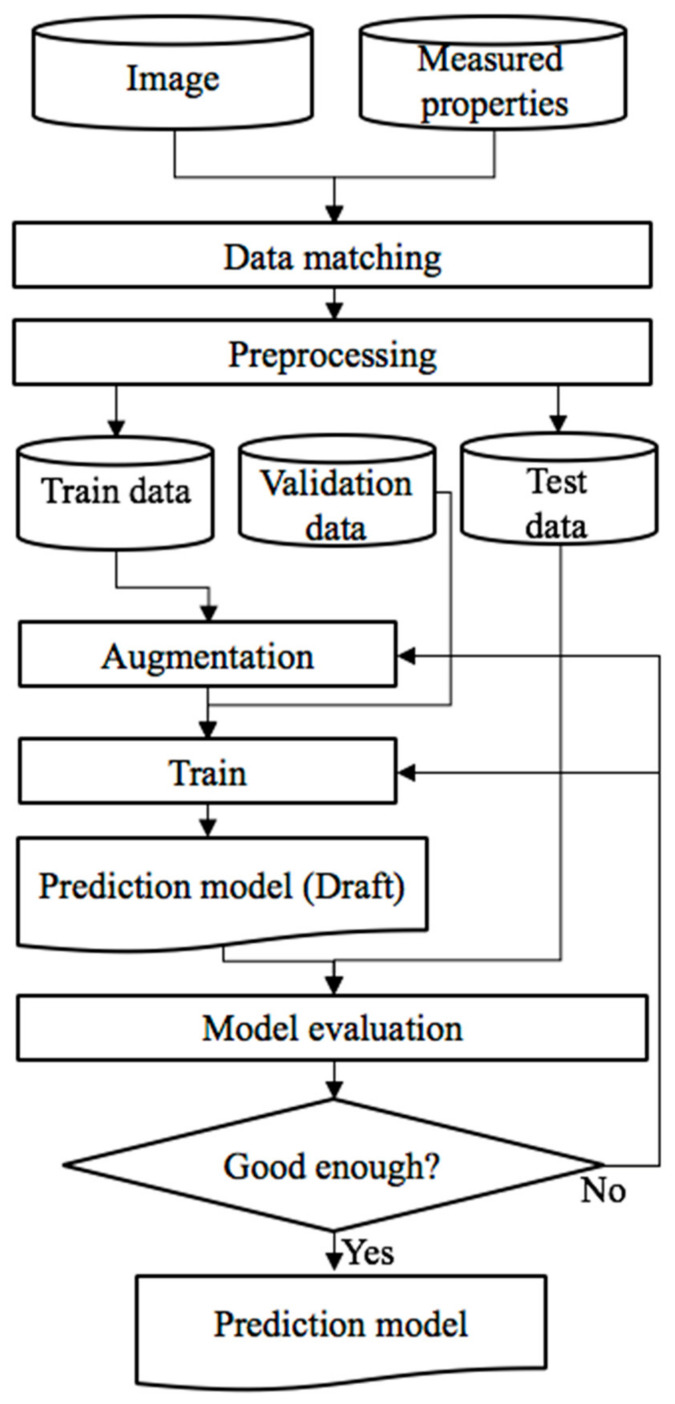
Flow chart of the proposed deep learning process.

**Figure 5 materials-13-05738-f005:**
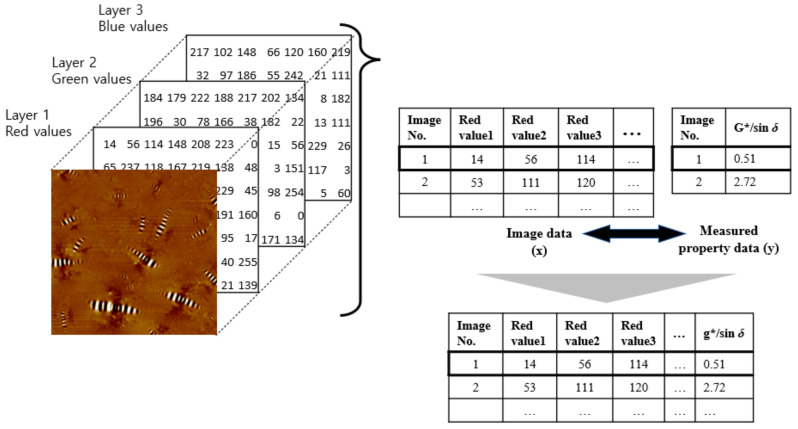
Conceptual diagram of data matching.

**Figure 6 materials-13-05738-f006:**
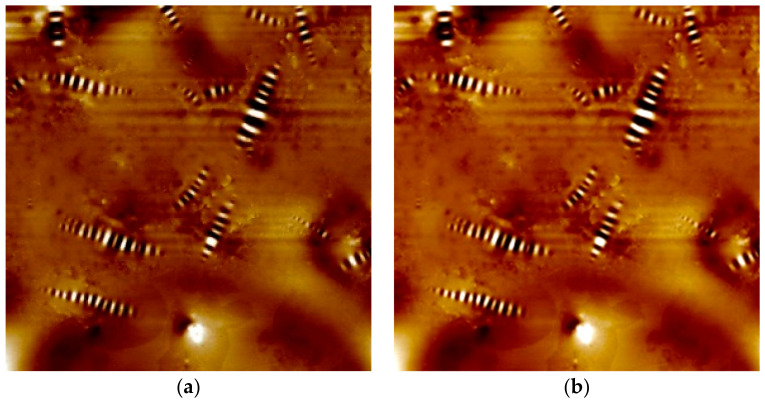
Color scale adjustment; (**a**) original image (**b**) color rescaled image.

**Figure 7 materials-13-05738-f007:**
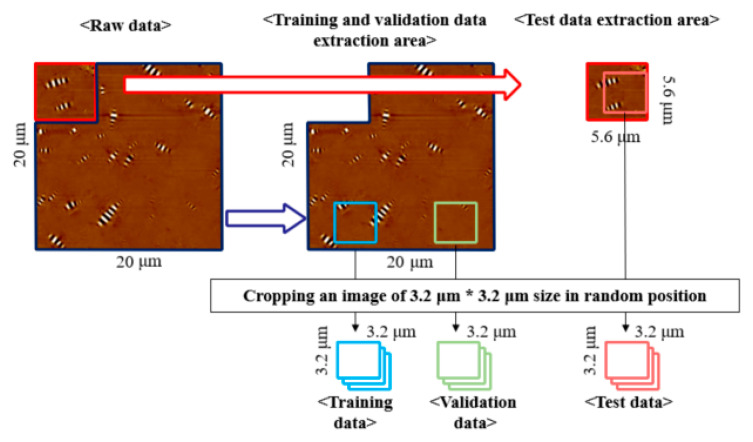
Image cropping process.

**Figure 8 materials-13-05738-f008:**
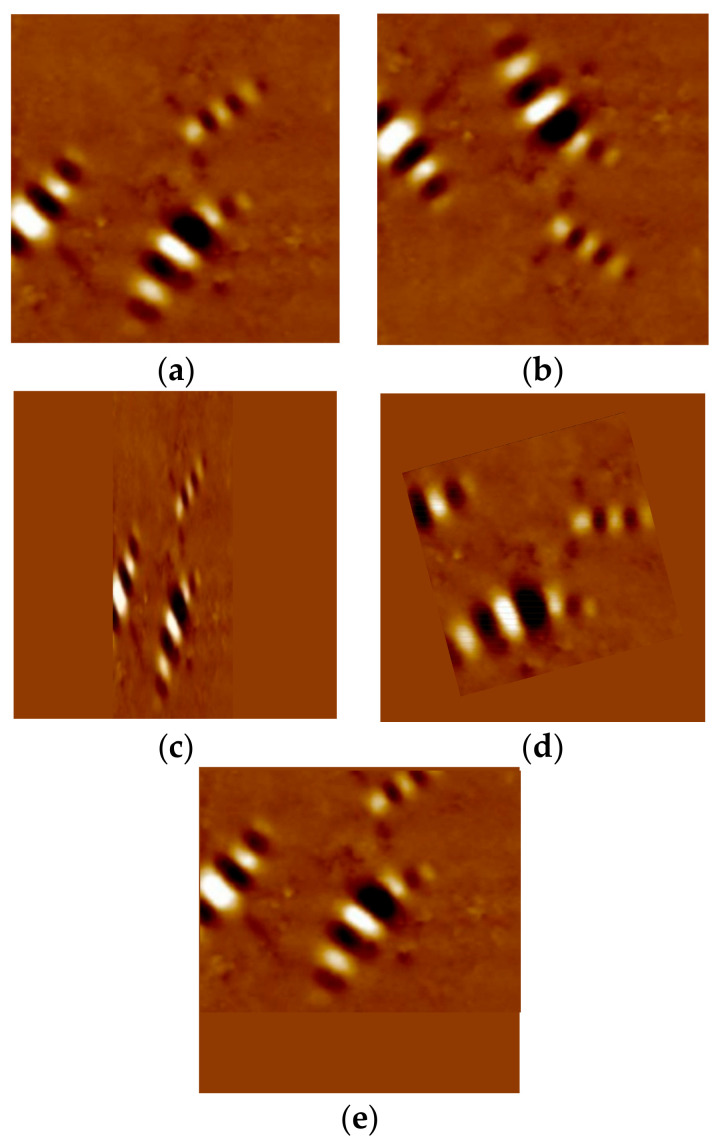
Examples of data augmentation; (**a**) original image, (**b**) flipping, (**c**) shearing, (**d**) rotating, (**e**) shifting.

**Figure 9 materials-13-05738-f009:**
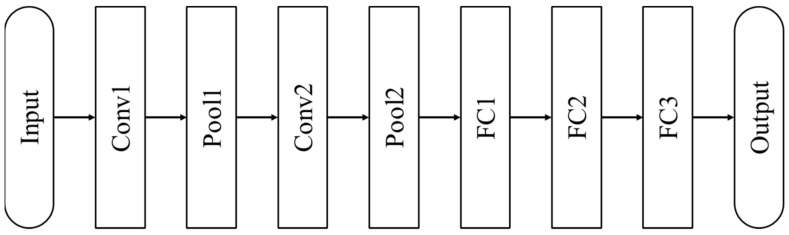
The LeNet-5 architecture [28].

**Figure 10 materials-13-05738-f010:**
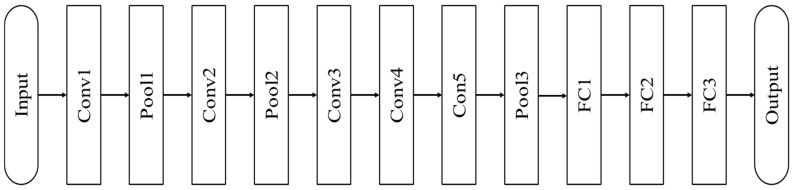
The AlexNet architecture [27].

**Figure 11 materials-13-05738-f011:**
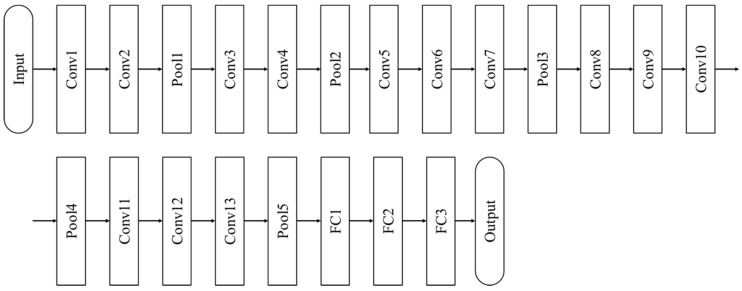
VGG-16 architecture [29].

**Figure 12 materials-13-05738-f012:**
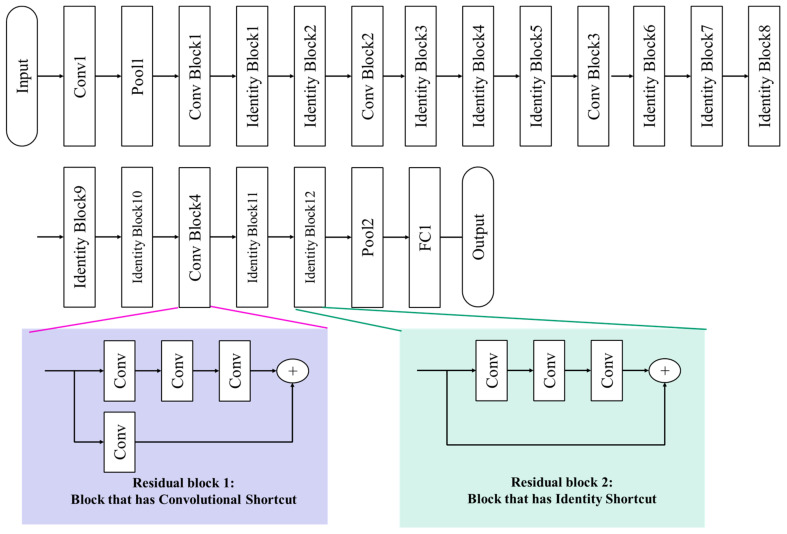
The ResNet architecture [25].

**Figure 13 materials-13-05738-f013:**
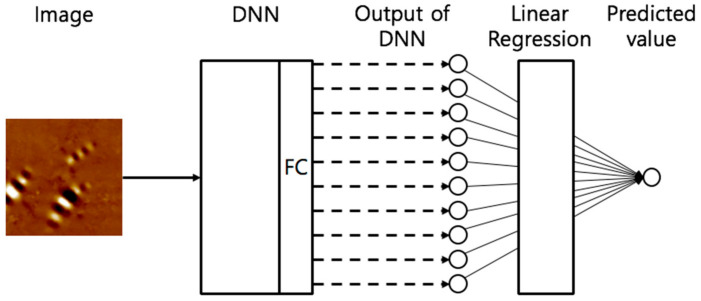
Conceptual architecture of group 2.

**Figure 14 materials-13-05738-f014:**
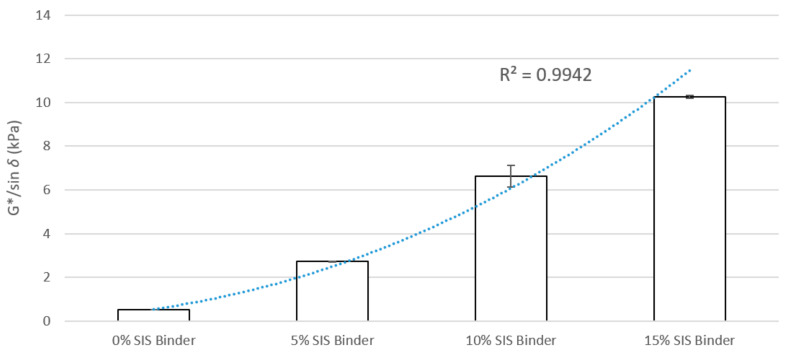
G*/sin *δ* at 76 °C (original).

**Figure 15 materials-13-05738-f015:**
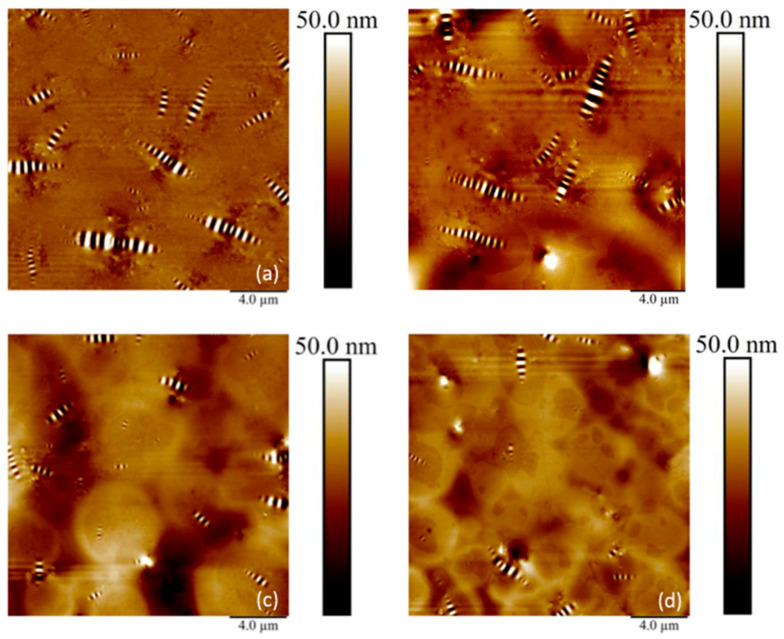
Atomic force microscopy (AFM) images of styrene–isoprene–styrene (SIS)-modified asphalt binders; (**a**) SIS 0% (**b**) SIS 5% (**c**) SIS 10% (**d**) SIS 15%.

**Figure 16 materials-13-05738-f016:**
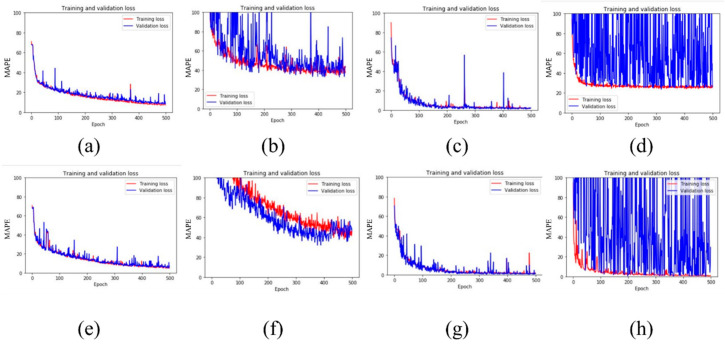
Training (red) and validation (blue) loss over 500 epochs; (**a**) LeNet5, (**b**) AlexNet, (**c**) VGG16, (**d**) ResNet50, (**e**) LeNet5+LR, (**f**) AlexNet+LR, (**g**) VGG16+LR, (**h**) ResNet50+LR.

**Figure 17 materials-13-05738-f017:**
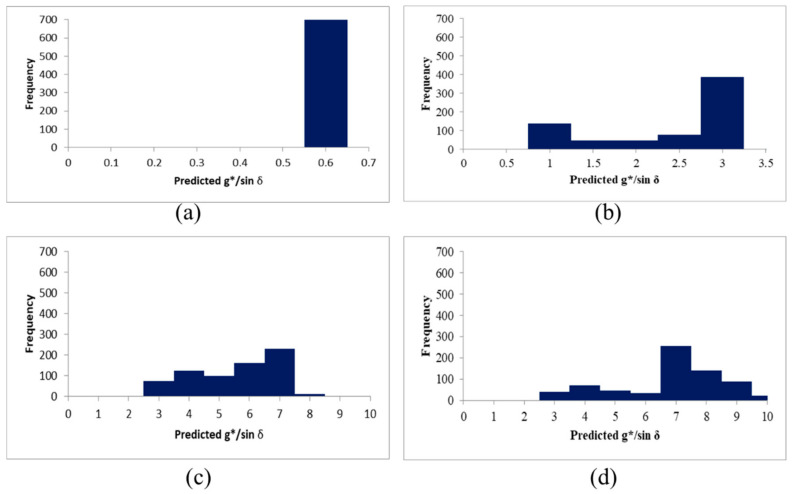
Distributions of the predicted value; (**a**) images whose actual value is 0.51, (**b**) images whose actual value is 2.72, (**c**) images whose actual value is 6.61, (**d**) images whose actual value is 10.25.

**Table 1 materials-13-05738-t001:** Properties of base asphalt binder (PG 64-22) [5].

Aging States	Test Properties	Test Result
Unaged binder	Viscosity at 135 °C (cP)	531
	G*/sin *δ* at 64 °C (kPa)	1.415
RTFO aged residual	G*/sin *δ* at 64 °C (kPa)	2.531
RTFO+PAV aged residual	G*sin *δ* at 25 °C (kPa)	2558
	Stiffness at −12 °C (MPa)	287
	m-value at −12 °C	0.307

**Table 2 materials-13-05738-t002:** Prediction results by eight applied architectures.

**Binder Type**	**Actual Value (G*/sin *δ*)**	**LeNet5**	**AlexNet**	**VGG16**	**ResNet50**
SIS 0%	0.51	0.52	0.00	0.51	0.00
SIS 5%	2.72	2.86	2.18	2.11	2.73
SIS 10%	6.61	3.24	4.36	5.13	6.58
SIS 15%	10.25	8.15	6.05	6.43	8.81
**Binder Type**	**Actual Value (G*/sin *δ*)**	**LeNet5+LR**	**AlexNet+LR**	**VGG16+LR**	**ResNet50+LR**
SIS 0%	0.51	0.51	0.40	0.51	0.51
SIS 5%	2.72	2.91	0.94	1.70	0.98
SIS 10%	6.61	3.78	3.24	3.37	6.70
SIS 15%	10.25	8.90	2.26	8.16	9.26

**Table 3 materials-13-05738-t003:** R^2^ of the prediction model for each architecture.

Architectures	R^2^	Rank
LeNet5	0.81	7
AlexNet	0.91	2
VGG16	0.88	4
ResNet50	0.96	1
LeNet5_LR	0.85	5
AlexNet_LR	0.59	8
VGG16_LR	0.85	5
ResNet50_LR	0.89	3

**Table 4 materials-13-05738-t004:** Mean squared error (MSE) of the prediction model for each architecture.

Actual Value	LeNet5	AlexNet	VGG16	ResNet50	LeNet5+LR	AlexNet+LR	VGG16+LR	ResNet50+LR
0.51	0.0	0.3	0.0	0.3	0.0	0.0	0.0	0.0
2.72	2.0	0.9	1.1	0.0	2.4	3.7	2.1	3.7
6.61	12.4	6.8	4.1	0.0	10.3	12.2	12.0	5.3
10.25	11.7	18.7	17.6	4.8	6.7	66.3	11.6	7.3
Average	6.5	6.7	5.7	1.3	4.9	20.6	6.4	4.1
Rank	6	7	4	1	3	8	5	2

**Table 5 materials-13-05738-t005:** Mean absolute percentage error (MAPE).

Actual Value	LeNet5	AlexNet	VGG16	ResNet50	LeNet5+LR	AlexNet+LR	VGG16+LR	ResNet50+LR
0.51	0.02	1.00	0.00	1.00	0.00	0.22	0.00	0.00
2.72	0.37	0.25	0.24	0.01	0.35	0.66	0.38	0.65
6.61	0.51	0.35	0.24	0.00	0.43	0.51	0.49	0.22
10.25	0.22	0.41	0.37	0.14	0.14	0.78	0.21	0.11
Average	0.28	0.50	0.21	0.29	0.23	0.54	0.27	0.25
Rank	5	7	1	6	2	8	4	3

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
