# Peer review of "Deep Regression Prediction of Rheological Properties of SIS-Modified Asphalt Binders"

_materials, 2020, doi:10.3390/ma13245738_

Round 1

Reviewer 1 Report

The topic of the paper is interesting. Unfortunately the merit of the article is undermined by several linguistic errors. Some of them and other comment are listed below.

Line 36
What do you mean by “through the elastic portion in material”? Do you mean that the elastic properties of polymer materials are able to overcome the weaknesses listed above? Please explain.

Line 59
“Even though the application of advanced technology positively supports to analyze the binder surface at a very small scale.” This is a fragment. Consider revising.

Figure 1
It appears that an arrow is missing between the first block of the third row and the first block of the fourth row.
“G*/sinδ”. Define all symbols the first time you use them.

Table 1
In column 2, the autocorrect has probably changed the word “at” to the “@” symbol. Please check.

Line 120
“The advantage of applying deep learning is that it excludes subjective views and provides results for objects.” For the sake of completeness, also list the disadvantages of deep learning. For example, anticipate here what you wrote in Section 2.4.6: “the performance of the deep learning model varies on the training data and hyper-parameter of the model”. Also list other disadvantages, if any.

Line 157
“Data augmentation, that is, the application of one or more variation to a collection of annotated training samples that generate new additional training data.” This is a fragment. Consider revising.

Line 196
“where the researchers who created the model”. Fragment. Complete the sentence.

Line 199
“VGGNet is consists of 5 blocks”. Delete “is”.

Line 201
“In this paper, we used VGG-16”. Explain why you chose to use this model. What makes VGG-16 preferable to the other models you listed?

Line 206
“The driving force behind ResNet's dramatic high performance is in layer depth lies in a module called Residual Block.” “The driving force … is in layer depth” or “The driving force … lies in a module called Residual Block”. Choose which of these two to use.

Line 219
“regression..”. delete one “.”.

Line 221
“label ,”. Delete the space before the comma.

Line 225
“Beacusre it is scale-independent metric.” Do you mean “Because”? Pay attention to fragments.

Line 228
“And Group 2 has an addition of linear regression layer to existing architecture (Figure 13).” Do not start a sentence with “And”.

Line 231
Delete the comma after “architecture”.

Line 249
“Additionally, based on the AFM image of SIS binders, it is required to observe how the results of the predicted values through the deep regression will show differences compared to the binder performance results obtained through the DSR test.” This sentence is not clear. Explain better what you mean by “differences”.

Line 267
“The further addition of SIS modifier with 15%”. Change “with” to “up to”.

Line 268
“The less number of bee structures in the topographic image, the trend is consistent with the earlier findings.” Check the sentence structure.

Line 272
“These new phases identified in Figure 15 (d), the oval shape appears to be brighter and smaller, covering more areas rather than previous image of SIS 10%.” Check the sentence structure.

Line 276
Delete the comma after “PG 64-22”.

Figure 15
Why is the label on the color bar (b) “60.0 nm” while the others are “50.0 nm”? Are the color scales actually different or is the label wrong?

Line 288

“… and will not enough performance to predict actual value”. You probably mean “perform”, not “performance”.

Line 307
“All architectures with linear regression showed lower R2 values rather than normal architectures except for the LeNet5.” The word “rather” does not seem appropriate in this sentence.

Line 316
“In this study, the smallest value is 0.51, and the largest value is 10.25, which can be a problem by more than 20 times difference.” Check the sentence structure.

Line 329
“For improved performance, more complex and advanced algorithms are applied, but it does not guarantee better performance.” This sentence is not clear. Explain better what you mean.

Line 333
Delete the comma after “analyzed”.

Line 358
Delete the comma after “images”.

Author Response

Dear Reviewer,

Thank you.

Reviewer 2 Report

The paper "Deep Regression Prediction of Rheological Properties of SIS Modified Asphalt Binders" studied regression model for predicting the engineering property of asphalt binders based on AFM image analysis. Here are my problems and questions:

* Why did authors use two architecture groups?

* Introduction is written simply; most recent research and innovation should be reviewed to show the gap of knowledge. Authors could use of the following works (synthesis unit) that relate to this work:

- Atomic force microscopy to investigate asphalt binders: a state-of-the-art review. Road Materials and Pavement Design 17, no. 3 (2016): 693-718.

- Ultraviolet aging study on bitumen modified by a composite of clay and fumed silica nanoparticles." Scientific Reports 10, no. 1 (2020): 1-17.

* Conclusion and abstract should be rewritten. The general information should be concisely. Instead, more details of reviewed aspects should be presented.

* The work is very same with previous paper of authors. Apparently, there are several same figures.
* Unit of Thermal insulation performance is shallow, and authors are expected to describe more details. Some assumptions are wrong.

* " DSR " section needs to be revised again and add empirical equations. Also need to present curves of complex modulus per temperature for comparison.

* Fig. 17 need to error bar.

* Validation of Prediction of rheological properties must be present for regression model.

Author Response

Dear Reviewer,

Thank you.

Round 2

Reviewer 1 Report

The reviewer appreciates the effort made by the authors to improve their manuscript. However, some further corrections are needed, as listed below.

Line 36 of the first document, now line 33
The reviewer appreciates the explanation given in lines 37-39. However, it is precisely the “elastic part in material” construct that does not make much sense. Rewrite this sentence with a different choice of words.

Answer A4
Since the quantity “G*/sind” also appears in Figure 1, you should move this explanation from Section 2.2. to Section 1.
Moreover, you call the complex modulus G, while in the formula you use G*. Is the right symbol to use for the complex modulus G* or is there a difference between complex modulus G and G*? In the latter case, explain the difference.

Answer A7
The sentence written in your answer is correct, but pay attention to the comma that has not been deleted in the text, after the word “is”.

Line 225 of the first document, now line 248
“Because it is scale-independent metric” is still a fragment.

Answer A16
The comma is actually still present in the new text.

Answer A22
There is still something that the reviewer fails to fully understand. If a scale adjustment was made in Section 2.4.2, why do we still have different color scales here in Section 3.2? Authors are encouraged to add some explanation to the color scales of Figure 15.

Answer A25
The two sentences in the new text are well structured. However, avoid putting a comma before the word “and”, unless it is the last item in a list with more than two terms. Please check and correct according to this rule throughout the text: this reviewer has not always indicated the comma to be deleted before the word “and”.

Author Response

Dear reviewer,

Thank you.

Reviewer 2 Report

Problems have been resolved.

Author Response

Dear reviewer,

We appreciate your contribution.

Thank you.